# Maternal Vaccination as an Integral Part of Life-Course Immunization: A Scoping Review of Uptake, Barriers, Facilitators, and Vaccine Hesitancy for Antenatal Vaccination in Ireland

**DOI:** 10.3390/vaccines13060557

**Published:** 2025-05-23

**Authors:** Adeyinka Sanni, Nuha Ibrahim, Dorothea Tilley, Sandra Bontha, Amy McMorrow, Roy K. Philip

**Affiliations:** 1Department of Public Health, University of Limerick School of Medicine, V94 T9PX Limerick, Ireland; 20175256@studentmail.ul.ie (A.S.); nuha.ibrahim@ul.ie (N.I.); thea.tilley@ul.ie (D.T.); sandra.bontha@studentmail.ul.ie (S.B.); 2Health Research Institute, University of Limerick, V94 T9PX Limerick, Ireland; 3Division of Neonatology, Department of Paediatrics, University Maternity Hospital Limerick (UMHL), V94 C566 Limerick, Ireland; 23329211@studentmail.ul.ie; 4University of Limerick School of Medicine, V94 T9PX Limerick, Ireland

**Keywords:** maternal vaccination, maternal immunization, life-course immunization, scoping review, vaccine hesitancy, antenatal care, pregnancy

## Abstract

**Background**: Maternal vaccination is a critical primary preventive approach and an integral part of life-course immunization strategy, influencing the infection-associated morbidity and mortality in pregnant women, foetuses, and young infants. Despite clear guidelines for the administration of vaccines against tetanus, diphtheria, pertussis (Tdap), influenza, and COVID-19 during pregnancy, maternal vaccination rates remain suboptimal in Ireland as per the National Immunisation Office of the Health Service Executive (HSE). **Aim:** This review explores the prevailing status, uptake factors, and maternal immunization-specific vaccine hesitancy in Ireland. **Method**: A scoping review was conducted, searching nine electronic databases, including the Irish health research repository Lenus. The search strategy utilised a Population–Concept–Context framework (pregnant women—vaccine uptake/hesitancy—Ireland). Key factors identified and categorised according to the 5A framework: access, affordability, awareness, acceptance, and activation. **Results**: Searches yielded 2457 articles, and 12 eligible studies were included for review. Influencing factors were identified in each of the 5A dimensions, with the majority relating to acceptance and awareness. Positively associated factors included healthcare provider (HCP) recommendation and knowledge of vaccine safety. Potential antenatal barriers were maternal lack of knowledge of vaccine-preventable illness severity, infection risks, and vaccine safety concerns. A pregnant woman’s primary motivation for antenatal immunization was protection of her infant; however, the reluctance of HCPs to prescribe all recommended antenatal vaccines, inadequate immunization-specific discussion during antenatal consultations, and suboptimal knowledge of pregnancy-specific vaccine safety hampered potential positive influences. The Irish national immunization policy was a facilitator of affordability. Activation can be achieved through public health awareness campaigns and interdisciplinary promotion of maternal vaccination uptake. **Conclusions**: Maternal vaccination uptake in Ireland remains suboptimal, and a coordinated, targeted approach updating HCP recommendations, enhancing maternal awareness, and highlighting vaccine safety in pregnancy would be required to meet the life-course immunization goals recommended by WHO. By adopting a life-course immunization approach for healthy living, with maternal vaccination as the pivotal central point, vaccination programmes could close immunity gaps at various life stages.

## 1. Introduction

Almost half of all child deaths occur in the first month of life [1], and 45% of under-five mortality is from infectious diseases [2]. For the mother–infant dyad, pregnancy and early infancy are vulnerable windows for infection [3]. The pregnant mother’s physiology and immunology undergo gestation-specific changes, increasing the risk of severe disease [4,5]. In parallel, the foetus develops an independent immune system, reaching relative maturity around three months after birth, making newborn infants highly susceptible to infection [6,7]. During this critical stage (and beyond), the maternal immune system provides passive protection to the foetus and newborn via the transfer of antibodies and antimicrobial proteins via placental circulation, and subsequently through colostrum and breast milk [8]. Colostrum is regarded as ‘*nature’s first vaccine*’, enriched with IgA antibodies and a multitude of bioactive properties [1,2]. Preterm infants pose additional challenges as they lack the opportunity for the trans-placental antibody (IgG) transfer that occurs in the last trimester [1]. Boosting immunity through maternal vaccination is a key strategy in protecting pregnant women and their infants against infectious diseases and reducing their severity, and is thus unique in providing *transgenerational primary prevention* through a single intervention [1].

Vaccination in pregnancy confers valuable passive protection to the newborn. Historically, the only vaccination offered in pregnancy globally was against neonatal tetanus (tetanus toxoid, TT) [3]. While the prevalence of neonatal tetanus has dramatically decreased around the world, it remains a preventable cause of death in low and middle-income countries (LMIC), posing a high mortality rate of 10–60% despite treatment [3,4]. More recently, low-dose diphtheria toxoid and acellular pertussis vaccine components were added, resulting in the Tdap maternal vaccine offering a triple prevention through one immunization strategy [5,6].

The WHO recommends maternal vaccination against respiratory pathogens, influenza, and pertussis (whooping cough) in all pregnancies [9,10]. More recently, COVID-19 is recommended on a risk-to-benefit basis [11]. The consequences of influenza in pregnancy extend beyond maternal health, from increased hospitalization due to cardiorespiratory complications to increased risk of stillbirth [12,13,14]. There is little evidence that influenza infects the foetus [15]; however, young infants (<6 months) have high hospitalization rates, prolonged stays in intensive care units, and increased mortality [16,17,18]. While pertussis can affect all age groups, pertussis-associated complications are more common among young infants [19,20]. Infants with pertussis often develop vomiting, cyanosis, apnoea, pneumonia, neurological complications, and sometimes death [21,22]. Furthermore, infections are easily transmitted from older children, mothers, and adults whose milder symptoms may go unnoticed [23,24]. While relatively new, COVID-19 during antenatal and perinatal periods is associated with increased risk of adverse outcomes, such as pre-eclampsia, preterm delivery, foetal loss, and neonatal SARS-CoV-2 (COVID-19) infection manifested by fever, pneumonia, and gastrointestinal symptoms [25].

More recently, the WHO and UNICEF recommended a global paradigm shift in the immunization strategy by adopting a *life-course approach* (LCA) to immunization, which involves vaccination programmes tailored to close immunity gaps at different stages of life. Maternal vaccination plays a pivotal role in this journey [26,27,28,29]. Maternal COVID-19 mRNA vaccination rollout and uptake over a relatively short timeframe during the pandemic could galvanise worldwide momentum and boost acceptance of the safety offered by antenatal vaccination, which occupies a strategic mid-position in the life-course vaccination spectrum from infancy to older adults [30]. Promotion of the interest and motivation engendered by the pandemic would be an opportunity to increase the adult vaccination rate globally, including vaccines recommended antenatally [30]. 

To support vaccine uptake, national organizations recommend specific vaccine protocols in pregnancy. In Ireland, the National Immunisation Advisory Committee (NIAC) of the Royal College of Physicians of Ireland (RCPI) advises the following: one dose of influenza vaccine at any time in pregnancy [31], one dose of Tdap vaccine between 16 and 36 weeks or any time after 36 weeks of pregnancy to maximise neonatal antibody level [32], and one course of COVID-19 mRNA vaccination at any point in pregnancy [33]. Both influenza vaccination and Tdap demonstrate no increased risk for adverse maternal or foetal outcomes [34,35,36,37]. However, vaccine coverage, disease surveillance, efficacy, and safety need continuous monitoring. This was highlighted in 2012 with a resurgence in pertussis cases globally [19], and Ireland grappled with a 301% increase in cases from 114 to 458 over a 2-year period [38,39]. Neonatal fatalities underscored the urgency of vaccination, with three deaths among unvaccinated infants <2 months of age [38,39]. Later, the 2015/2016 flu season was characterised by a resurgence in the pandemic H1N1 strain, and its associated hospitalizations underscored the urgency of maternal immunization [40].

Despite clear recommendations, maternal vaccination rates in Ireland remain low [38]. However, the uptake of childhood vaccines included in the National Immunization Programme (Appendix A) is satisfactory despite the recent challenges posed by the COVID-19 pandemic. Previous studies shed light on factors affecting maternal vaccination [41]. This scoping review aims to map and summarise factors influencing maternal vaccination in Ireland, employing Thomson et al.’s 5A framework, a taxonomy of vaccine determinants: access, affordability, awareness, acceptance, and activation [42]. This framework focuses on the potential root causes of vaccine coverage gaps, outside of sociodemographic factors. By examining these, we can identify barriers and facilitators related to maternal vaccination in Ireland, ultimately informing targeted interventions to improve vaccination uptake among pregnant women. This review fills a gap in the existing literature, offering valuable insights into the specific context of maternal vaccination in Ireland. Observations from this review would also have a transferable relevance to optimise maternal immunization in other countries. 

## 2. Methods

### 2.1. Study Design

This review uses a scoping methodology [43,44]. The process for conducting the review was divided into five stages: identification of the research question; identification of relevant studies; selection of relevant studies; charting the data; and collating, summarising, and reporting the results. The conduct and reporting of results were further guided by the Preferred Reporting Items for Systematic reviews and Meta-Analyses (PRISMA) Extension for Scoping Reviews [45].

### 2.2. Statement of Research Question

This scoping review maps the available literature on maternal vaccination in Ireland and aims to answer: “What factors contribute to the acceptance or refusal of maternal vaccination in Ireland?”

### 2.3. Identification of Relevant Studies

To identify relevant articles, a search of multiple electronic databases was performed during the period March 2023 to May 2023: Embase; EBSCO (Academic Search Complete, CINAHL, APA PsycInfo, and Medline); PubMed; Scopus; Web of Science; and the Irish health research repository—Lenus. The search strategy utilised a Population–Concept–Context (PCC) framework, with the population being pregnant women, the concept being vaccine uptake/hesitancy, and the context being Ireland. The keywords used for the search strategy included terms related to pregnant women, vaccination, hesitancy/uptake, and Ireland. The detailed search strategy (Appendix A) and an example optimised for PubMed are described in the Appendix A. Searches were conducted from the inception of the databases, with the exception of Lenus, where search results were limited to 2013 to 2023 and search terms were restricted (Appendix A). An English language limitation was applied, and the reference lists of selected articles were hand-searched to identify additional relevant studies. 

### 2.4. Selection of Relevant Studies

To enable screening and selection of relevant studies, search results were imported into Endnote^®^ citation manager and then exported into Rayyan^®^, an online screening platform for systematic reviews [46]. After removing duplicates, the screening process involved title and abstract screening, followed by a full-text assessment by two independent reviewers (AS, SB) applying inclusion and exclusion criteria outlined in Table 1. Where only an abstract was available, authors were contacted by email to establish if a full-length article was available. If not, it was excluded. Any discrepancies were resolved through discussion with senior authors (NI, RKP). 

### 2.5. Charting the Data

Study data was extracted and charted in by two of the authors (AS and SB) using Excel as follows: title; author; year of publication; study aim/purpose; study design/methodology; population and sample size; location of study; vaccine type; and factors associated with uptake/hesitancy. To facilitate the identification and extraction of factors encompassed by Thomson et al.’s 5A taxonomy (Appendix A) [42], qualitative analysis software, Nvivo^®^, was used. Thomson’s 5A framework was used as the coding framework. The definitions of the 5A dimensions are as follows: ‘access’—the ability of individuals to be reached by, or to reach, recommended vaccines; ‘affordability’—the ability of individuals to afford vaccination, both in terms of financial and non-financial costs (e.g., time); ‘awareness’—the degree to which individuals have knowledge of the need for, and availability of, recommended vaccines and their objective benefits and risks; ‘acceptance’—the degree to which individuals accept, question, or refuse vaccination; ‘activation’—the degree to which individuals are nudged towards vaccination uptake [42]. Articles were coded to these dimensions. For multinational studies, results relevant/specific to the Irish population were extracted where possible.

### 2.6. Collating, Summarizing, and Reporting the Results

Information from selected studies was collated and summarised in tables, and results of the screening and selection process were summarised using the PRISMA flow diagram [47]. A narrative synthesis was prepared to report the findings. Risk-of-bias assessment and quality of evidence of the eligible studies were not envisaged as part of the scoping review. 

## 3. Results

Searches of nine electronic databases yielded 2458 results, with one article found through a hand search. Among these, 865 references were sourced from the Irish health research repository, Lenus, while 43 were obtained from PubMed. After screening, the authors of four conference abstracts that were not found as full texts were contacted and confirmed that one was never published, and another formed part of a study already included. The final number of articles included in the review was 12 (Figure 1).

### 3.1. Study Characteristics

Among the twelve studies selected for review, seven were quantitative, two qualitative, and three used mixed methods. There were four cohort studies and two cross-sectional studies. Two studies were connected, as the sampled participants were from the same cohort. Participants in the studies included pregnant women (6/12—50% of studies), post-partum women or breastfeeding women (3/12—25%), hospital and community health care providers (HCPs) (3/12—25%), and one study linked to the congenital anomaly register. Participants were recruited at maternity hospitals in 25% of studies, antenatal clinics associated with maternity hospitals (17%), at GP clinics (17%), and from the general community via online and in-person market research approaches (25%). Six studies (50%) examined both influenza and pertussis vaccination, three studies (25%) focused on influenza, one on pertussis vaccination alone, and two studies on willingness to receive the COVID-19 vaccine. Studies were distributed nationally, with four conducted in Dublin, one national study covering the Republic of Ireland, one reported from Northern Ireland in the Belfast region, and two international comparative studies (one covering 6 European countries and the other analysing data from 24, including Ireland) (Appendix A).

### 3.2. Sociodemographic Characteristics Associated with Maternal Vaccination

Overall, the reported vaccination uptake rates in pregnant or post-partum women ranged from 40 to 55.1% for influenza and 31 to 67% for Tdap vaccination [48,49,50]. Sociodemographic characteristics statistically associated with vaccine uptake were maternal age >30 [48,51,52,53] and higher socioeconomic status [48,51,54]. Higher educational attainment was associated with better influenza vaccination in pregnancy [48]. One study reported vaccination was more common in those identifying as Irish descent, while those women originally from Eastern Europe, Africa, and Asia/Middle East were less likely to be vaccinated [51]. During the 2017/18 Influenza season in Ireland, 241 pregnant women were enrolled in a cross-sectional survey, yielding an influenza and pertussis vaccine uptake of 61.7% and 49.9%, respectively [54].

### 3.3. The 5A Factors Associated with Maternal Vaccination

Applying Thomson’s 5A taxonomy, the proportion of studies reporting factors associated with maternal vaccination uptake or hesitancy in the 5A dimensions were: 10/12 (83%) acceptance, 5/12 (42%) access, 7/12 (58%) affordability, 9/12 (75%) awareness, and 3/12 (25%) activation (Appendix A). 

### 3.4. Acceptance

#### 3.4.1. Individual Characteristics

Individual characteristics associated with maternal vaccination were pregnancy status, trust, and health beliefs. Historically, antenatal vaccination uptake (excluding tetanus toxoid on a global scale) remained suboptimal [54]. Those more likely to report being vaccinated or willing to receive a vaccine were in their first pregnancy (primigravida), at gestational stage >20 weeks, had their first birth (primiparous), or were breastfeeding [52,53,55]. In contrast, unplanned pregnancy was associated with reduced uptake of the influenza vaccine [51]. In general, women conveyed trust in their HCP, knowledge, and advice [52,56,57]. However, a qualitative study demonstrated that a lack of consistency in care providers made it difficult to build trust [57]. The same study revealed a bias towards medication avoidance during pregnancy extending to immunizations as well [57]. This latter view was also held by some GPs in a study conducted in the West of Ireland [49].

#### 3.4.2. Perception of Illness 

One quantitative study showed that a higher proportion of pregnant women perceived that pertussis or influenza infection would be more dangerous for the infant than themselves [56]. Another study of COVID-19 vaccine uptake showed 5278/6661 (86%) of pregnant women believed severe COVID-19 during pregnancy could affect foetal development [55]. This concern for the safety of the infant was paramount and also arose in qualitative findings [52,57]. Another study demonstrated most women (157/198—79%) agreed they were more vulnerable to illness during pregnancy than when not pregnant [48]. Among the surveyed HCPs, 883/1180 (74.8%) agreed that influenza in pregnancy increased the risk of foetal complications. However, only 615/1180 (52.1%) identified that it could lead to preterm delivery, miscarriage, and foetal death [48]. 

#### 3.4.3. Social Context of HCPs

HCPs’ influence was a key determinant of maternal immunization uptake in multiple studies [50,52,54,56,57,58]. A recommendation from a GP was the most common reason for vaccine uptake in two studies [48,54], and in another, inadequate information from HCPs was the most common reason for non-uptake [50]. Qualitative studies revealed a lack of consistency in how regularly or intently HCPs discussed and recommended vaccination. Some vaccinated women highlighted HCPs’ lack of engagement and multidisciplinary coordination [54,58] and the need for more vaccine-related discussion during appointments [57]. Some unvaccinated women reported they were neither offered a vaccine nor discussed its merits or safety with any HCP but would accept the vaccine if recommended [51,57,58]. Others perceived the vaccine was not important in pregnancy, if not endorsed by HCPs [57]. In one quantitative study, 54/113 (48%) of HCPs answered they never discussed pertussis vaccination with pregnant patients [50]. 

Limited peer influence was observed in the reviewed studies. Friends and family were considered sources of information [50,57]; however, their influence was not considered a major factor in decision making [58]. Social and family responsibility coupled with ‘uptake conflict’ was represented in pregnant women’s priority to protect the infant from perceived adverse effects [48,52,57]. One study noted that protection of infants from influenza was the most common reason for uptake [48]. Pregnant women felt it was more important to protect the baby than themselves, but did not understand vaccination as a means to protect the infant via passive immunization [57]. In parallel, GPs more than other HCPs felt it was their responsibility and role to recommend (101/109—92.9%) and offer (100/109—91.7%) the vaccine [48], and female HCPs were more likely to do so [49]. 

#### 3.4.4. Vaccine

Concern about vaccine safety was a common reason for HCPs choosing not to recommend vaccination [48,49,50,52,54]. In one study, 81% of GPs who did not recommend pertussis vaccination had safety concerns versus 29% of recommending GPs [49]. HCPs’ safety concerns predominantly related to insufficient data in pregnancy and the potential for future complications [49]. Maternal concern about vaccine-mediated harm to the foetus was paramount and associated with reduced uptake [48,52]. Two qualitative studies revealed that concerns about vaccine safety derived from a perceived risk of vaccination and development of long-term conditions (narcolepsy and autism), despite this being disproven [51,57,58]. Conversely, knowing a vaccine was safe was associated with uptake or uptake intent [50]. 

Negative perceived efficacy arose as a factor in one qualitative study where pregnant women reported that they knew someone who received the flu vaccine but also developed flu [54]. However, women who had received a vaccine in a previous pregnancy or had been vaccinated for influenza were more likely to be vaccinated for pertussis, indicating a positive attitude to vaccination [55,58].

### 3.5. Affordability

No studies explicitly examined the relationship between vaccine affordability in Ireland and vaccine uptake. However, four studies observed that maternal vaccination uptake was influenced by public or private funding of antenatal care. Women receiving publicly funded care were less likely to receive or express intent to accept vaccination compared to women receiving privately funded care [48,51,52,53]. In an international comparative study, public policy on funding of vaccination during pregnancy differed between countries [59]. Currently, in Ireland, there is no patient-level cost for vaccinations included in the national immunization schedule, although previously an administration fee was applied [49,60]. Fewer than 20% of GPs felt women were declining the pertussis vaccine due to financial cost [49]. Following a national vaccine funding policy change triggered by a pertussis outbreak, vaccine administration fees were removed in an effort to increase uptake [54,61]. One qualitative study reported lack of time, affording unpaid time off work, and obtaining childcare as barriers to vaccination [57]. 

### 3.6. Access

#### 3.6.1. Convenience

GPs were reported as most commonly administering vaccines, and women praised vaccine availability at community GP visits [48,54,58]. Vaccines were also delivered by pharmacists, midwives and hospital doctors, occupational health services, and nurses in GP clinics [54,58]. Notably, a qualitative study revealed that some women felt having to attend a GP specifically for vaccination was inconvenient, and vaccines could be administered by midwives during antenatal appointments [57]. A quantitative study of HCPs’ attitudes to vaccination revealed that the majority of pharmacists and hospital doctors were of the view that vaccination should be administered in any location [48]. A contemporary study of pregnant women indicated that antenatal pertussis vaccine was not discussed sufficiently in the hospital environment [58]. 

#### 3.6.2. Location of Vaccination

One study noted differences in uptake of maternal vaccination according to location, with pregnant women living in the western (Connacht) or northern (Ulster) provinces less likely to be vaccinated than those in the east (Leinster) [54]. The same study showed a greater uptake of influenza vaccine in rural areas, but an inverted relationship with pertussis vaccination [54].

#### 3.6.3. Contact with Healthcare Systems

Two pregnancy-related factors, unplanned pregnancy and late booking (gestational stage >20 weeks at the first booking visit with antenatal care), were associated with reduced likelihood of being vaccinated for influenza [51]. These factors may be viewed as reasons and evidence for less contact with the health system during the course of pregnancy, and possibly less opportunity to access information on vaccination. However, being in the later stages of gestation (>20 weeks) was also associated with increased willingness to receive the COVID-19 vaccine [62]. 

### 3.7. Awareness

Factors relating to the degree of awareness and knowledge individuals have of vaccination were identified in multiple studies and are categorised by the pregnant women’s or HCPs’ awareness/knowledge.

#### 3.7.1. Women’s Knowledge of Vaccines and Vaccine Schedule

Three studies examined pregnant women’s awareness and knowledge of vaccination [50,57,58]. Two identified most pregnant women were aware of vaccination, irrespective of intent [50,57]. Specifically for pertussis vaccination, a qualitative study revealed that 7/17 post-partum women had not heard of antenatal pertussis vaccination [58]. In another study, over half the participants were unsure if there were risks associated with influenza vaccination in pregnancy [58]. In contrast, a quantitative study revealed that pregnant women’s knowledge of either influenza or pertussis vaccine safety in pregnancy was associated with better uptake [50]. 

#### 3.7.2. HCPs’ Knowledge of Vaccines and Vaccine Schedule

Three studies examined HCPs’ awareness and knowledge of vaccination in pregnancy, revealing high overall awareness but varying levels of knowledge on specific guidelines [48,49,50]. One single site study of non-consultant hospital doctors (NCHDs) and midwives found 95% were aware of the national recommendations for influenza and pertussis vaccination during pregnancy [50]. One national online survey, mostly of GPs and pharmacists, found 97.6% knew influenza vaccination was recommended, but only 75.6% knew about pertussis vaccine recommendation [48]. Furthermore, for NCHDs and midwives, the majority lacked specific knowledge of the schedule for pertussis vaccination, specifically its reduced effectiveness if given in the last weeks of pregnancy and its recommendation for unvaccinated women within one week of birth [50]. 

#### 3.7.3. Availability of Information

GPs were the most common sources of information and considered reliable, but obstetricians, hospital doctors, midwives, antenatal classes, and healthcare settings in general were also common sources of information [48,50,54,56,57,58]. Information was provided in the form of discussions and also through public health information leaflets [54]. A qualitative study identified that women felt leaflets in the absence of discussion were not helpful [57]. Outside of healthcare settings, information was obtained through friends and family, social media, websites, and literature [48,54,57,58], and pregnant women compared information from different sources with the public health information [26]. With respect to COVID-19 vaccination, pregnant women valued scientific information, but there were also differences in the interpretation of data [52]. 

### 3.8. Activation

#### 3.8.1. Prompts and Reminders

One study noted that during a pertussis outbreak, letters were sent to GPs and public health nurses, possibly mediating the observed increase in vaccination [54]. Crucially, structured educational programmes and support for HCPs were recommended to address HCP vaccine confidence and hesitancy to recommend, and to enable effective communication with patients for shared decision making [48,49,50,52,55,56]. To initiate a direct vaccine activation of pregnant women, GPs recommend that a pertussis vaccination reminder be included in maternity notes and antenatal information packs [49,56]. 

#### 3.8.2. Workplace Policies

Vaccination for influenza and pertussis is recommended for HCPs working with infants in Ireland and should be offered by employers [63]. HCP uptake of vaccines was examined in two studies. Both revealed that many HCPs still chose not to be vaccinated [48,50]. The majority (76%) of NCHDs and midwives had not been vaccinated against seasonal influenza at the time of study and did not intend to receive it [50]. In contrast, in an online survey of HCPs (432 GPs, 417 pharmacists, and 29 hospital clinical staff), only a minority (267/878—30.4%) had not received the vaccine [48]. 

## 4. Discussion

This scoping review reaffirms the primacy of pregnant women’s sense of responsibility and intent to protect their unborn infant, and the positive influence of HCPs as the major influencing factor in maternal vaccination uptake. HCP recommendations are positively associated with uptake, but their delivery—in terms of frequency and sufficiency of information provided—is lacking. Effective communication of vaccine risks and benefits is hampered, complicating pregnant women’s decision to be vaccinated due to unassuaged concerns for vaccine safety for herself and even more for the unborn infant. The reticence of some healthcare providers to recommend vaccination is attributed to concerns about potential vaccine adverse effects and is exacerbated by a lack of support from the hierarchy. This remains a formidable hurdle. Pregnant women seek sources of information outside of healthcare settings to inform their decisions, indicating a wider potential sphere of influence, and public awareness campaigns have proved effective, overcoming established sociodemographic barriers to vaccination acceptance in Ireland. This is particularly relevant in the post-COVID-19 era by countering misinformation in a timely and effective manner, cultivating a culture of antenatal vaccine acceptance through targeted antenatal education of women and HCPs, and ensuring equitable access through interdisciplinary coordination [26,30]. 

Healthcare professionals wield substantial social influence in shaping vaccine confidence among pregnant women, as identified in a recent meta-analysis [64]. However, a divergence of views within the healthcare community regarding vaccine safety highlights the need for comprehensive safety data to bolster providers’ confidence. To the authors’ knowledge, there is no routine monitoring of pregnancy-specific vaccine safety indicators, and a robust system for the timely dissemination of the collated information to the HCPs involved in maternal immunization in Ireland, thus hampering HCPs decisions to recommend and also leaving them ill-equipped to answer questions on vaccine safety from pregnant mothers, compounding maternal vaccine hesitancy.

In the reviewed studies, neither access nor affordability arose as notable barriers to maternal immunization in an Irish context, while vaccination programmes among sub-populations in some European regions were reported to be negatively influenced by patient-level and provider-level affordability concerns [65]. However, in the majority of the included studies, pregnant or post-partum women were recruited at sites of access, such as GP clinics, or potential access, such as maternity hospitals, and so this was unlikely to be a barrier for these participants. Expanding access by widening who can provide vaccines, specifically antenatal services and pharmacies, has been shown to increase uptake [66,67,68]. Questions around accessibility, beyond GP clinics, remain, and previous studies have identified time, storage issues, and availability of staff certified to administer vaccines as challenges to access [69,70]. Affordability was not an issue due to free provision under Irish policy; however, it is unclear why privately funded obstetric care is associated with better pregnancy-specific vaccine uptake compared to publicly funded obstetric care. This association is found in studies from the US, Europe, and Australia, as recently reviewed by Geoghegan et al. [52]. In the US, this is thought to be due, in part, to high demand on public services, HCP decisions according to insurance coverage, and HCP-perceived ability to pay [71].

A significant gap was observed in provider training and support that is intensifying vaccine hesitancy among pregnant women. For women, this combination of shortfalls may create an omission bias where women choose no action rather than a potential yet improbable harm to their infant [42]. Similar factors were identified in a scoping review by MacDougall et al. [72]. Future research should establish the informational needs of pregnant women and complementary HCP training. Increasing knowledge and awareness of the importance of maternal vaccination and its efficacy are important strategies for increasing vaccination coverage [26,73]. 

Further to the completion of this scoping review, the largest antenatal survey of pregnant women in Europe on awareness, acceptability, and willingness to receive RSV maternal vaccination was published from Ireland [74]. Of 528 women, a large proportion (75.6%) had never heard of RSV, yet 48.5% would still avail themselves of a vaccine, 45.8% were undecided, and only 5.3% would not. The main factor making vaccination acceptable to women (76.4%) was protection of their infant from illness (*p* < 0.001, CV 0.336 for association with acceptance) [74]. Since the completion of our data synthesis, one scoping review and one qualitative systematic review on related aspects of maternal immunization highlighted women’s need for clear information, ideally provided by their healthcare professionals, in a meaningful way that addresses identified barriers [75,76]. Governments, regulatory authorities, and HCPs have to collectively educate pregnant women about the effectiveness and safety of maternal vaccines and encourage vaccination when the benefits outweigh the risks [77]. Consistent and evidence-enabled recommendations from HCPs, with mindfulness of the sociodemographic and cultural background of pregnant women, would foster the acceptance of vaccination in pregnancy as an essential component of antenatal care. Global experience and lessons learned from the rollout of the COVID-19 vaccination during the antenatal period provided valuable insights that can inform current and future approaches to optimise maternal vaccination strategies [30,78]. 

## 5. Limitations

Thomson’s 5A taxonomy is based on factors supported by evidential statements, and we conducted this scoping review in a similar manner, examining each study for factors influencing or related to the maternal vaccine coverage landscape in Ireland. However, due to the limited number of quantitative studies and study designs, the majority of factors identified, while often supported by qualitative findings, do not represent statistical associations with vaccine uptake. Additionally, unlike in a systematic review, no critical appraisal of the literature was conducted to determine the rigour and validity of the findings, and no risk-of-bias analysis and quality of evidence assessments were performed as part of this scoping review. Therefore, care is recommended when deciding which factors to target in future intervention designs to optimise the antenatal vaccination rates as an integral part of a life-course preventative approach [26,29,30].

## 6. Conclusions

This analysis, within a 5A framework, illuminates the intricate web of factors that shape maternal vaccination uptake in Ireland. From knowledge gaps to healthcare provider perspectives, risk perceptions, and safety considerations, the narrative encompasses a diverse range of determinants that are likely responsible for the low maternal vaccination rates observed. Potential effective strategies for improving antenatal vaccination uptake are consistent and sustained recommendations by HCPs, including reminders in antenatal care packs and targeted public awareness campaigns, along with optimal utilization of social media. By addressing the identified barriers and harnessing facilitators, a roadmap can be charted towards a landscape where maternal vaccination is embraced as a critical part of a life-course immunization strategy, fostering the well-being of both mothers and infants. It is paramount to recognize that national immunization programmes need realignment and redesign with inter-stakeholder and inter-agency participation to ensure comprehensive protection for individuals across the lifespan. By adopting a life-course approach to immunization, with maternal vaccination as the pivotal central point, vaccination programmes could close immunity gaps at various life stages. 

## Figures and Tables

**Figure 1 vaccines-13-00557-f001:**
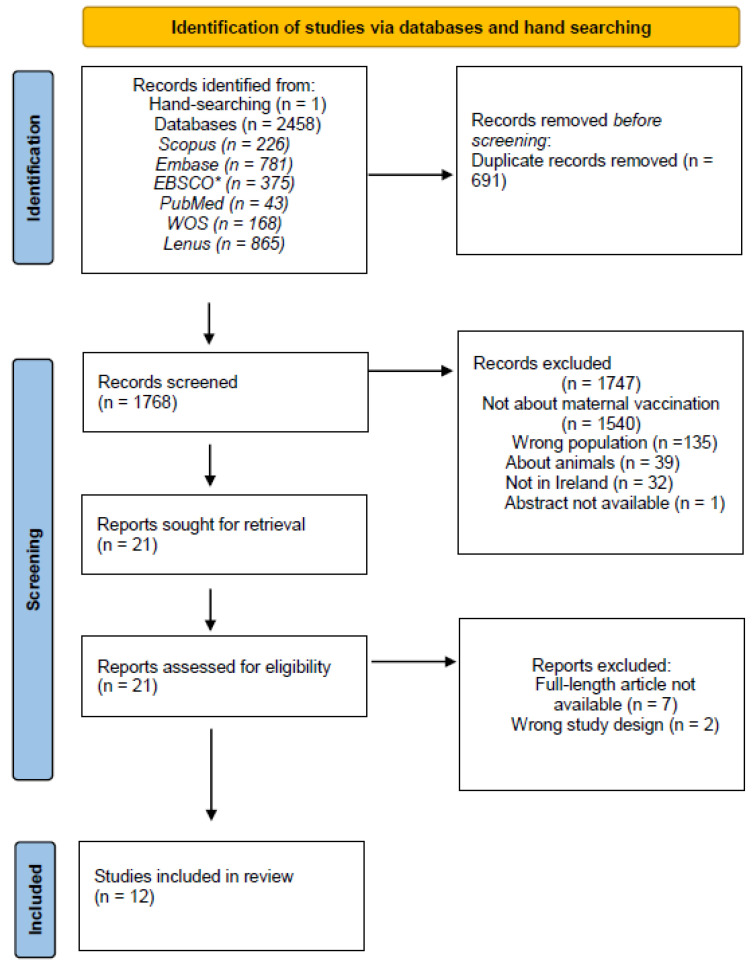
PRISMA flow diagram reporting search results, screening, and exclusion decisions. WOS—Web of Science. * EBSCO search of 4 databases: Academic Search Complete, CINAHL, APA PsycInfo, and Medline.

**Table 1 vaccines-13-00557-t001:** Inclusion and exclusion criteria.

Inclusion Criteria	Exclusion Criteria
Population: pregnant women or women of reproductive age	Studies conducted outside of Ireland.
Concept: factors influencing maternal vaccination	Studies focusing solely on non-pregnant women.
Context: Studies conducted in Ireland.	Animal studies, editorials, opinions, commentaries.
Publication types: peer-reviewed articles, freely accessible articles.	Studies not written in English.
Methodological approaches: quantitative, qualitative, mixed-method studies.	Studies with insufficient data or inadequate reporting.

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
