# Peer review of "Maternal Vaccination as an Integral Part of Life-Course Immunization: A Scoping Review of Uptake, Barriers, Facilitators, and Vaccine Hesitancy for Antenatal Vaccination in Ireland"

_vaccines, 2025, doi:10.3390/vaccines13060557_

Round 1

Reviewer 1 Report

Comments and Suggestions for Authors

·         Add a link to Ref 9 - 71

·         Despite clear recommendations, maternal vaccination rates in Ireland remain low === Provide your statement with a suitable reference

·         References are not correctly organized and numbered.

·         There are some typo errors

·         The review should be provided with a figure.

·         I think that it is better to integrate the supplementary data into the main manuscript.

·         Vaccines used in Ireland should be mentioned in detail, like company, dose,...

·         The official performed programs of vaccinations in Ireland

·         Can you mention cases of vaccination program failure in Ireland?

·         Can you provide WHO vaccine coverage regarding Ireland?

·         Is there universal health coverage (UHC)? – if yes, mention its free services to women

Comments on the Quality of English Language

English should be improved

Reviewer 2 Report

Comments and Suggestions for Authors

Although this review comprehensively explored maternal vaccination uptake in Ireland, the study has some gaps that it could have addressed. There is a need for critical appraisal or an assessment of the risk of bias in order to enhance the dependability of the studies that were included. The review also requires a deeper probe on the gaps in the training among health service providers and systemic barriers to shed light on actionable strategies for improvement on the vaccine recommendation rates. Additionally, deeper explorations of sociodemographic factors, such as ethnicity and socioeconomic status, would help provide a nuanced look at vaccination disparities. Comparative analysis with other high-income countries on Ireland's policies regarding immunization would put the findings into context and inform best practices. The disparities in rural-urban access to vaccines, and consideration of non-traditional settings for vaccination, like pharmacies and community programs, would improve vaccine access. Also, making coordination a point between GPs, midwives, and public health would offer consistent messaging with the intention of coordinating vaccine delivery, thus increasing the vaccination uptake rates.

Reviewer 3 Report

Comments and Suggestions for Authors

Dear Authors,

Your study is very interesting. Its results are important for public health in Ireland and other countries as well. I have several comments.

Table 1: it should not be a picture. Please change it to an editable table.

Supplementary materials are very useful but please pay attention that two tables are named S4.

Reviewer 4 Report

Comments and Suggestions for Authors

Line 18. Could you provide the statistical analysis methods used?

Line 32. ….Maternal vaccination uptake in Ireland is suboptimal

Comment: Could you show that in the results?

Line 106. …..Ireland remain low

Comment: Provide the reference

Lines 176-177. This is a repletion paragraph. I suggest to be deleted

Results.

General comments.

o  Provide information on persons implement and did the identification of relevant studies, chatting the data,  etc. Are they independent?

o  Provide n=? Whenever you provide % ie 25%;n=?

Line 203 and 229 …..pertussis vaccination. Could add (Tdap)

General comments: Are they any studies looking at postCOVID-19 pandemic uptake of these vaccines?

Line 227. Diseases. I suggest to have a more clear title rather than just a word diseases

Line 237. I suggest to read Social context of HCP

Line 440. The conclusion is a little lengthy; I suggest that you focus and shorten it.

Round 2

Reviewer 2 Report

Comments and Suggestions for Authors

Could be accepted in the current form.